# Label-free detection of conformational changes in switchable DNA nanostructures with microwave microfluidics

Angela C. Stelson[1], Minghui Liu[2,3], Charles A.E. Little[1], Christian J. Long[1], Nathan D. Orloff[1], Nicholas Stephanopoulos [2,3] & James C. Booth[1]

Detection of conformational changes in biomolecular assemblies provides critical information into biological and self-assembly processes. State-of-the-art in situ biomolecular conformation detection techniques rely on fluorescent labels or protein-specific binding agents to signal conformational changes. Here, we present an on-chip, label-free technique to detect conformational changes in a DNA nanomechanical tweezer structure with microwave microfluidics. We measure the electromagnetic properties of suspended DNA tweezer solutions from 50 kHz to 110 GHz and directly detect two distinct conformations of the structures. We develop a physical model to describe the electrical properties of the tweezers, and correlate model parameters to conformational changes. The strongest indicator for conformational changes in DNA tweezers are the ionic conductivity, while shifts in the magnitude of the cooperative water relaxation indicate the addition of fuel strands used to open the tweezer. Microwave microfluidic detection of conformational changes is a generalizable, non-destructive technique, making it attractive for high-throughput measurements.

---

[1] National Institute of Standards and Technology, Radio Frequency Electronics Group, Boulder CO 325 Broadway St, Boulder, CO 80305, USA. [2] School of Molecular Sciences, Arizona State University, 551 E University Dr, Tempe, AZ 85281, USA. [3] Center for Molecular Design and Biomimetics, The Biodesign Institute, Arizona State University, 727 E. Tyler St., Tempe, AZ 85281, USA. Correspondence and requests for materials should be addressed to N.S. (email: nstepha1@asu.edu) or to J.C.B. (email: james.booth@nist.gov)

Detecting conformational changes in large biomolecular assemblies, such as proteins, protein complexes, and DNA origami is critical to understanding the function of these systems. Standard characterization methods to detect nanoscale changes in biological systems include cryogenic transmission electron microscopy (cryo-TEM) and atomic force microscopy (AFM)[1,2]. These techniques, however, require extensive and destructive sample preparation (cryo-TEM, for example), are limited in spatial resolution (AFM, for example), and can be perturbative to the state in situ. They also typically capture the one-end state of the conformational switch, making real-time measurements of intermediate conformations technically challenging and limited to single molecules[3].

A common technique to probe biomolecular dynamics in situ is Förster resonance energy transfer (FRET), where the distance between two fluorophore labels is calculated from the energy transfer efficiency between the fluorophores. The limit of FRET-based distance measurements is around 1–10 nm (depending on the dye pair) and requires site-specific modification of the targeted analyte with the two fluorophores[4]. Site modifications are technically challenging, and the size and hydrophobic nature of the dyes make them potentially perturbative[5]. To address these limitations, label-free methods to probe conformational changes are an active area of research. These label-free methods also detect protein binding with protein-specific binding agents (e.g., aptamers and antibodies)[6]. Typically, these protein-specific binding agents are attached to a surface, and the ligand association perturbs a measurement signal. Ligand association detection techniques include impedance spectroscopy, surface-enhanced Raman scattering (SERS), acoustic waves, and calorimetry[7–13]. However, both label- and aptamer-based protein detection techniques require prior knowledge of the analyte and the targeted experimental design[14].

A label-free, generalizable method of detecting molecular conformation changes in solution will advance high-throughput and real-time biological characterization. Compared with expensive and time-intensive alternatives (e.g., FRET), on-chip, high-throughput methods facilitate quantitative characterization at potentially lower costs. Orthogonal techniques for detecting conformation changes in protein-based systems would also allow for cross-comparisons, for example, to confirm that fluorescent labels do not interfere with biomolecular dynamics in FRET measurements. Fluorescent label interference inhibits progress in "nano-machine" engineering, requiring characterization techniques that do not rely on surfaces or labels to detect conformational changes[14,15].

Frequency-dependent dielectric measurements can detect the structure and dynamics of proteins, DNA, and cells in solution without the need for specific binding agents[16–24]. Recent advances in microwave metrology and instrumentation allow for on-chip broadband dielectric measurements from DC to 110 GHz[25–29]. Covering this wide-frequency range allows for measurements of electrical properties of different charge-based phenomena in a system that includes electrical double layers, ionic conductivity, and molecular reorientations[26,30–33]. Microwave microfluidics is an emerging field of study that integrates microfluidics with on-chip microwave devices and electrical measurement techniques, allowing for quantitative measurements of nanoliter volumes of fluids, and provides a probe mechanism of aqueous and ionic solutions, particles, cells, and more[34–39].

Here, we utilize microwave microfluidics to track the conformational changes in DNA "tweezer" nanostructures. In and of itself, the DNA tweezer system provides a wide array of biological detection capabilities, and similar structures have been used to measure protein–protein interactions[40,41]. By using a nanostructure that is well-defined with controllable binary states, these microwave microfluidics experiments elucidate the relevant electrical properties of the fluid that signal conformational changes. This ability to detect DNA nanostructure changes electrically means that we can use these nanostructures as a model system to study large conformational changes in similar systems. In addition, we can modify tweezers to detect and amplify small conformation changes in complex biological systems. The former application will be useful when FRET dyes or other probes are intractable due to the difficulties with synthesis or excessive perturbation to the complex biological system under test. In the latter application, DNA tweezers can serve as a model system that mimics conformational changes in protein mechanisms[42] and DNA origami[43,44]. The electrical characterization methods developed here could provide a more user-friendly and high-throughput alternative to optically based measurements, such as FRET[45].

The microwave microfluidic techniques we develop here are widely applicable to any biological fluid system, opening avenues for high-throughput in situ measurements for use by the biotechnology, molecular biology, and biomanufacturing communities. Microfluidics allow integration with complementary lab-on-a-chip technologies, including optical, thermal, mechanical, and chemical stimuli and measurements, with high throughput and nanoliter sample size[46,47]. Our devices and calibration protocol cover a six-decade frequency bandwidth from 50 kHz to 110 GHz. This large bandwidth uniquely captures both the low-frequency regime where the electrical-double-layer effects dominate as well as the high-frequency regime where the properties of the solution dominate. We determined the presence of a weak relaxation associated with the ion pairing in solution by fitting the entire frequency regime of the dielectric spectrum on a logarithmic scale. We extracted quantitative model parameters associated with the ion-pairing relaxations, the bulk fluid properties, and the electrical double layer, and correlated these parameters to DNA tweezer conformational changes.

## Results

**DNA tweezers**. As a model system for probing conformational changes, we chose a DNA tweezer nanostructure[48] consisting of two rigid arms held closed by a hairpin stem–loop and two extended locking strands (Fig. 1a, sequence in Fig. 1; see Supplementary Information). The tweezer can be opened by the addition of two fuel strands: one that binds to the hairpin (FUEL 1) and the other that breaks the locking strands (FUEL 2). FUEL 1 forms a duplex that rapidly forces the arms apart, in a spring-loaded fashion. We can trigger a large conformational change in the tweezer between a closed state (4-nm inter-arm distance) and an open state (16-nm inter-arm distance). The clean transition from closed to open tweezers with addition of fuel strands was confirmed by atomic force microscopy (AFM, Fig. 1b–d) and native polyacrylamide gel electrophoresis (PAGE, Fig. 1e). As a control, we added dummy fuel strands (Fig. 1a) that were mismatched to their targets, which resulted in no change in tweezer conformation (Supplementary Figures 2 and 3). We hypothesized that the large difference between the closed and open states of the tweezer would be detectable in the frequency shift of a relaxation associated with the tweezer in solution, a phenomenon that has been reported for DNA and protein suspensions[20,33].

**The microwave microfluidics device**. We developed a microwave microfluidics device (Fig. 2a) to measure the broadband electrical properties of suspensions of DNA tweezers. The device consisted of microfluidic channels with integrated coplanar waveguides (CPWs) of varying lengths. Each of these devices (top–down view in Fig. 2b) was connected to a vector network analyzer (VNA) via

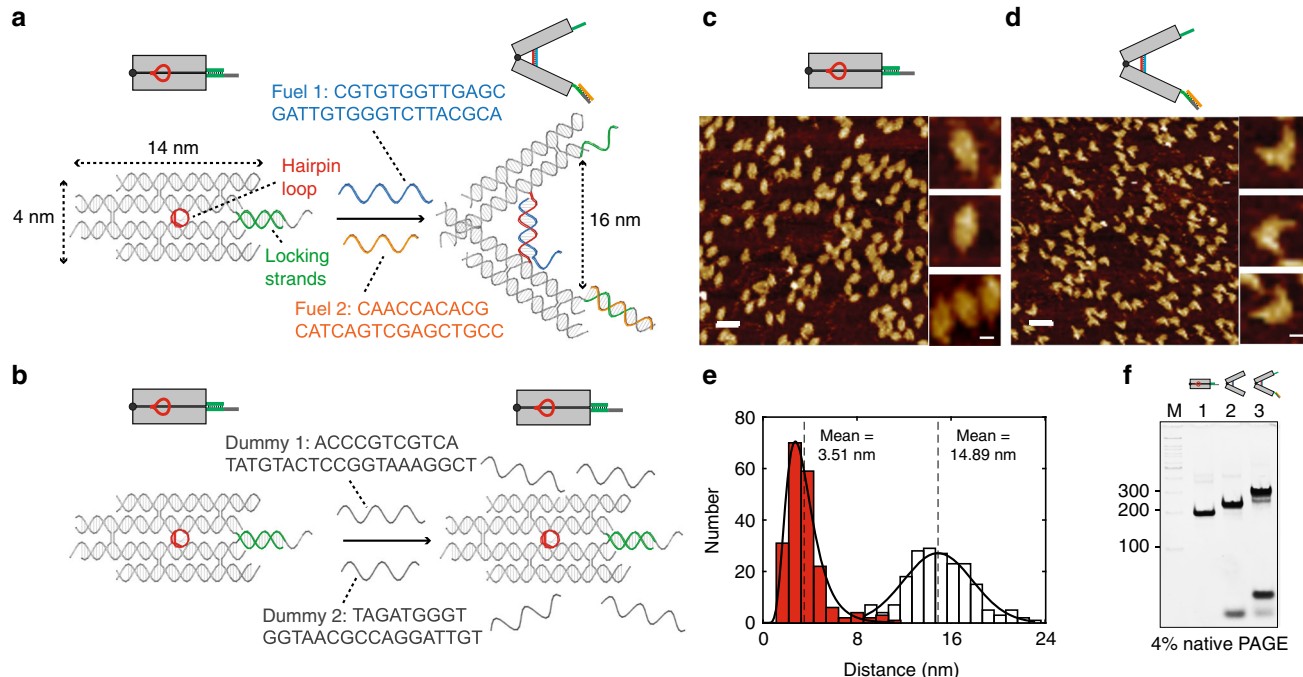

**Fig. 1** Design of DNA tweezers and characterization of closed and open tweezers. **a** Design and dimensions of a closed tweezer with locking strands (green) and an open tweezer with addition of fuel strand 1 (FUEL 1, blue) and fuel strand 2 (FUEL 2, orange). **b** Control tweezers with dummy fuel strands 1 and 2 (gray). **c** Zoom-out and zoom-in AFM images of closed tweezers with locking strands. **d** Zoom-out and zoom-in AFM images of open tweezers after addition of both fuel strands. Scale bars are 50 nm for zoom-out images and 10 nm for zoom-in images. **e** Histograms for the distance between the ends of the tweezer arms based on AFM imaging (red and white histograms are separate samples of closed and open tweezers, respectively). **f** Native PAGE gel characterization of tweezer opening: lane M, double-stranded DNA ladder as a standard marker (distance along the gel marked in units of base pairs); lane 1, closed tweezer with locking strands; lane 2, open tweezer with addition of FUEL 1; lane 3, open tweezer with both FUEL 1 and FUEL 2

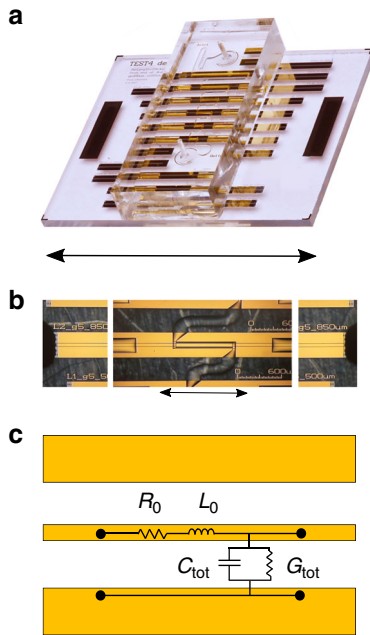

**Fig. 2** Microwave microfluidics devices and circuit schematic. **a** Image of the microwave microfluidics device (scale bar is 12 mm). **b** Composite microscope image of microfluidic channels with microwave probes landed (scale bar is 1 mm). **c** Circuit model that describes the electrical behavior of the CPW. The distributed circuit parameters $R_0$, $L_0$, $C_{tot}$, and $G_{tot}$ are frequency-dependent per unit length quantities, and $C_{tot}$ and $G_{tot}$ depend on the fluid properties

microwave probes to measure the raw scattering parameters (S parameters) as a function of frequency. The S-parameters were calibrated (see the Methods section) and used to extract the distributed circuit parameters of the transmission line $R_0$, $L_0$, $C_{tot}$, and $G_{tot}$, which correspond to the resistance and inductance associated with the metal conductors in the transmission line, and the capacitance and conductance associated with the materials in the gap (Fig. 2c), respectively. We present calibrated fluid data as $C_{tot}$ and $\frac{G_{tot}}{\omega}$ as these quantities can be related to the real and imaginary parts of the fluid permittivity ($\varepsilon'$ and $\varepsilon''$), respectively:

$$\varepsilon' = (C_{tot} - C_{air})k + \varepsilon_0 \text{ and} \quad (1)$$

$$\varepsilon'' = \frac{G_{tot}}{\omega}k, \quad (2)$$

where $C_{air}$ is the per-unit-length capacitance of an air-filled channel, $\varepsilon_0$ is the permittivity of free space, and $k$ is a geometric constant dictated by device structure[26]. While Eqs. (1) and (2) allow us to convert capacitance $C_{tot}$ and scaled conductance $\frac{G_{tot}}{\omega}$ directly to fluid permittivity values, we do not present measured data in terms of permittivity when electrical double layer (EDL) effects are present, because the EDL effects depend on the device geometry and are not directly related to intrinsic fluid properties.

**Fluid measurement and circuit model.** We measured the distributed conductance and capacitance of air, deionized water (DI water), tris-acetate-ethylenediaminetetraacetic acid with magnesium chloride (TAE-$Mg^{2+}$) buffer, closed tweezers, and open tweezers (0.5 μM concentration, both suspended in TAE-$Mg^{2+}$ buffer) (Fig. 3a, b). The capacitance of air remained constant as a function of frequency, while the DI water had a larger capacitance

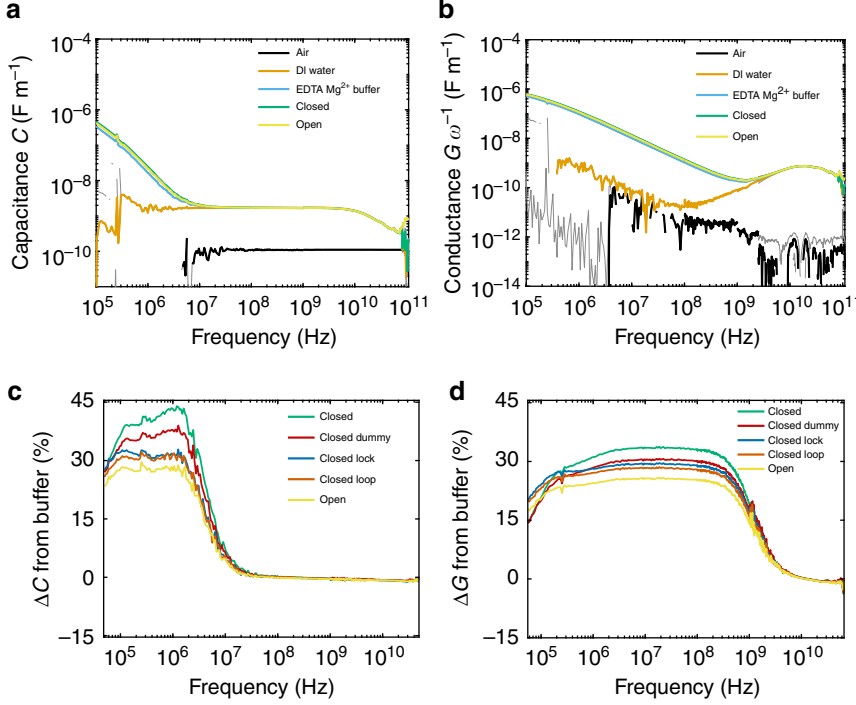

**Fig. 3** Calibrated **a** distributed capacitance $C_{tot}$ and **b** distributed conductance $G_{tot}\,\omega^{-1}$ for air, DI water, TAE-$Mg^{2+}$ buffer, closed DNA tweezers, and open DNA tweezers. Lighter gray lines represent 95% confidence intervals (shown on all datasets). The percent change from TAE-$Mg^{2+}$ buffer for closed, open, and control tweezer samples for **c** distributed capacitance $C_{tot}$ and **d** distributed conductance $G_{tot}\,\omega^{-1}$. Loop and Lock correspond to the additions of FUEL 1 and FUEL 2, respectively

at low frequencies and a relaxation (a peak in the conductance paired with a drop in the capacitance) at ~20 GHz due to the cooperative relaxation of water molecules[49]. This water loss peak was present in the aqueous solution samples as well. Previous dielectric spectroscopy studies reported weak relaxations for proteins (β- and δ-relaxations) and electrolytes (ion-pairing relaxations) at frequencies in the range of 10 MHz to 1 GHz[24,50]. However, these relaxations were approximately three orders of magnitude smaller than the water relaxation, and required careful fitting treatment to extract quantitative information from dielectric data. At low frequencies (below $10^7$ Hz), we saw a peak in the conductance and a drop in the capacitance for the solutions containing ions, which we attributed to the relaxation of the EDL. To clarify the changes in the broadband electrical properties for DNA tweezers, we plotted the percent deviation in $C_{tot}$ and $G_{tot}\,\omega^{-1}$ from the TAE-$Mg^{2+}$ buffer as a function of frequency (Fig. 3c, d) for open and closed tweezers, as well as control measurements with dummy strands of DNA, and tweezers with only FUEL 1 or FUEL 2 added (~5 μM concentration), labeled as Lock and Loop, respectively, in Fig. 3c, d. We observed an increase in $C_{tot}$ and $G_{tot}$ upon the addition of tweezers, and a reduction upon the addition of single-stranded DNA (fuel strands or dummy fuel strands, Fig. 1a). Open tweezers had values for $C_{tot}$ and $G_{tot}$ that were smaller than both the closed tweezer and all the control samples.

To extract physical values from the broadband electrical data, we developed a circuit model to describe total admittance (inverse of impedance) $Y_{tot} = G_{tot} + i\omega C_{tot}$ of the suspended DNA tweezers (Fig. 4):

$$\frac{1}{Y_{tot}} = \frac{2}{Y_{EDL}} + \frac{1}{Y_f},\tag{3}$$

where $Y_{EDL}$ and $Y_f$ are the admittances of the EDL and fluid, respectively. We describe the effect of the EDL as operating in series with the admittance of the fluid for fluids with dissolved

ions. The EDL can be modeled as a Cole–Cole relaxation:[26]

$$Y_{EDL} = Y_{CPE} + G_{EDL} + i\omega C_{EDL} = Y_{CPE} + i\omega\frac{C_{EDL}}{1 + (i\omega\tau_{EDL})^{1-\alpha_{EDL}}},\tag{4}$$

where $C_{EDL}$ is the capacitance associated with the EDL, $\alpha_{EDL}$ is a shape-broadening parameter, and $\tau_{EDL}$ is the characteristic relaxation time associated with the formation of the EDL under an electric field. The Cole–Cole relaxation is in parallel with a constant-phase element (CPE, with admittance $Y_{CPE}$):

$$Y_{CPE} = Q\omega^{-n}e^{i\frac{\pi}{2}n},\tag{5}$$

where $Q$ and $n$ are the fitting parameters, and where $Q$ has the units [S $m^{-1}$ $Hz^n$]. We fixed $n = -1$ since the concentration of ions in the sample was low, and allowing $n$ to vary did not change the fit parameters.

We described the fluid admittance $Y_f$ as four parallelly distributed circuit components:

$$Y_f = Y_{IP} + Y_w + G_\sigma + i\omega C_\infty = i\omega\frac{C_{IP}}{1 + (i\omega\tau_{IP})} + i\omega\frac{C_w - C_\infty}{1 + (i\omega\tau_w)} + G_\sigma + i\omega C_\infty,\tag{6}$$

where $C_\infty$ is the capacitance of the suspension at frequencies far above the relaxation frequency of water, $C_w$ is the dipolar contribution of the water, $G_\sigma$ is the conductance due to translation of ions and DNA, and $C_{IP}$ is the dipolar contribution of the weak ion/DNA relaxation. The values for $G_{ions}$ and $G_w$ (Fig. 4a) represent the loss (imaginary part) of the Debye relaxations $Y_w$ and $Y_{IP}$ and are not separate fitting parameters. The time constants $\tau_w$ and $\tau_{IP}$ correspond to the rotational relaxation times of the water and the ion–counterion pair, respectively. By developing an equivalent circuit model based on Debye-type relaxations, we correlated the changes in charge-based phenomena to the changes in DNA tweezer conformation,

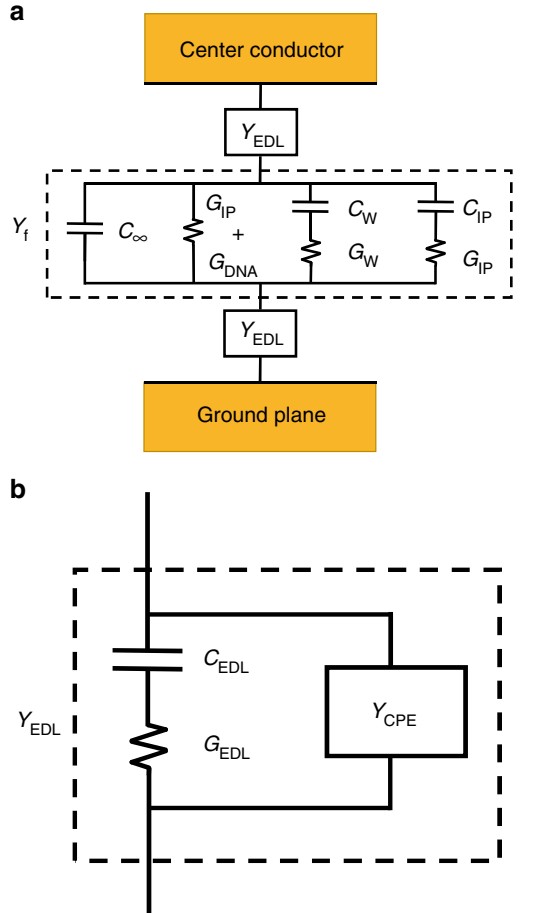

**Fig. 4** Circuit model for DNA tweezers suspended in TAE-Mg$^{2+}$ buffer. **a** Total fluid admittance $Y_{tot}$ is comprising the EDL admittance $Y_{EDL}$ in series with $Y_f$. **b** Equivalent circuit of $Y_{EDL}$, where $Y_{CPE}$ is the admittance of the constant-phase element (CPE), and $C_{EDL}$ and $G_{EDL}$ represent the Debye relaxation form of the EDL

using our broadband measurements. Here, we determined statistical significance in the changes in fit using a two-sided z test with unequal variance and applying the Bonferroni criterion for two simultaneous statistical tests within a single dataset[51].

**Fitting the fluid data to fluid circuit parameters**. To fit these regions with Debye-type models, we performed a nonlinear least-squares fit to extract $Y_{tot}$. Specifically, we simultaneously fit log ($C_{tot}$) and log($G_{tot}$) (Fig. 5a, e) for the whole frequency spectrum, and $C_w$ and $G_w + G_\sigma$ for the frequency range 5–30 GHz, using fit parameters and functions normalized to have magnitudes approximately equal to one. The inclusion of $C_w$ and $G_w + G_\sigma$ into the fitting model at high frequency was necessary to address the colinearity between $G_\sigma$ and CPE effects in the model, which constrained CPE effects to lower frequencies. The full frequency range of the fit was necessary to achieve the uncertainties presented here for all extracted fit parameters. The bulk fluid properties contain the water relaxation $Y_w$ as well as the fluid conductance $G_\sigma$, and the fit and the corresponding data are presented in Fig. 5b, f. Including the ion relaxation peak ($Y_{IP}$ in Fig. 5c, g) was necessary in DNA tweezer suspensions and control measurements of TAE-Mg$^{2+}$ buffer to produce symmetric (Cole–Cole) relaxations for the EDL, and resulted in overall lower residuals across the high-frequency regime. Residuals for fits for a single Cole–Cole relaxation versus two Debye-type relaxations at

high frequency have been reported elsewhere for Mg$^{2+}$–EDTA buffer, and are included for closed tweezers in Fig. 3[52].

**Detecting conformational changes**. We tracked fit parameters from the $C_{tot}$ fit for the TAE-Mg$^{2+}$ buffer, closed tweezers, open tweezers, and a series of control measurements. The controls included closed tweezers with dummy strands that do not open the tweezers, and closed tweezers with FUEL 1 or FUEL 2 added. By extracting quantitative information from our calibrated broadband dielectric measurements, we determined physical parameters that strongly indicate ($p < 0.05$, two-sided Z test with unequal variance) conformational changes in the tweezer system. The value $C_w$ (Fig. 6a) corresponds to the dipolar contribution of the cooperative water relaxation, and indicates changes in the state of the water. Reduction in the dipolar contribution of water can come from displaced water molecules, water immobilized on the surface of the DNA tweezers (i.e., bound water in hydration layers), changes in the concentration-dependent charge density of DNA, and changes in the charge state of the buffer[20,32,53,54]. All tweezer samples had smaller $C_w$ values, and there was a statistically significant increase in $C_w$ from both closed and closed-with-dummy-strands to the open configuration ($p < 0.05$, two-sided Z test with unequal variance). To determine the impact of excess fuel strands, we varied the amount of excess fuel strand in the open tweezer sample from 4.5 to 7.5 μM (see Supplementary Figure 8). The linear relationship between $C_w$ and the concentration of fuel strands suggests that the change in $C_w$ is due to the additional single strands of DNA in the solution, rather than a change in conformation. Notably, the magnitude of $C_w$ increased upon the addition of more single-stranded DNA, meaning that there is less bound water in the system, overall. This counter-intuitive result could be due to the changes in the buffer, or the concentration-dependent charge density of the DNA itself[53,54].

The relaxation time of the water loss peak $\tau_w$ also shifts when DNA or ions are added to water (Fig. 6b). Shifts in the water relaxation in ionic, protein, and DNA systems have been attributed to disruption of the hydrogen-bonding network and increases in solution viscosity[55]. An increase in $\tau_w$ occurred for the buffer, and the addition of closed tweezers further increased the relaxation time. Open tweezers and tweezers with dummy strands had a reduced water relaxation time, compared with closed tweezers, demonstrating that single-stranded DNA can cause a reduction in $\tau_w$ similar to what we observe in the open tweezers. However, when we varied the excess FUEL strands in open tweezers, we found that changing the concentration of excess FUEL strands did not affect $\tau_w$ (Supplementary Figure 9). This suggests that $\tau_w$ could be an indicator of conformational changes in the tweezers, and the FUEL strands and dummy have distinct effects on the water relaxation. In addition, studies of globular proteins in water found a linear relationship between $C_w$ and $\tau_w$. This finding was not the case in the DNA tweezer system, which we attributed to the presence of diverse, complex charged species in solution[55].

The bulk ionic conductivity $G_\sigma$ is a sensitive indicator of both the addition of fuel strands as well as the conformational changes of the tweezers (Fig. 6c). The closed tweezers had a greater ionic conductivity compared with the buffer, consistent with molecular orbital theory and measurements of DNA systems with nanopores and dielectric spectroscopy[20,56–59]. The addition of dummy FUEL strands slightly decreased the overall ionic conductivity of the solution, suggesting a lower ionic conductivity for single strands in solution, as compared with tweezer structures. We observed reductions in $G_\sigma$ beyond the dummy strand control measurements for the addition of the locker and central loop strands (FUEL 1 and FUEL 2, respectively), demonstrating that these binding events are detected on an

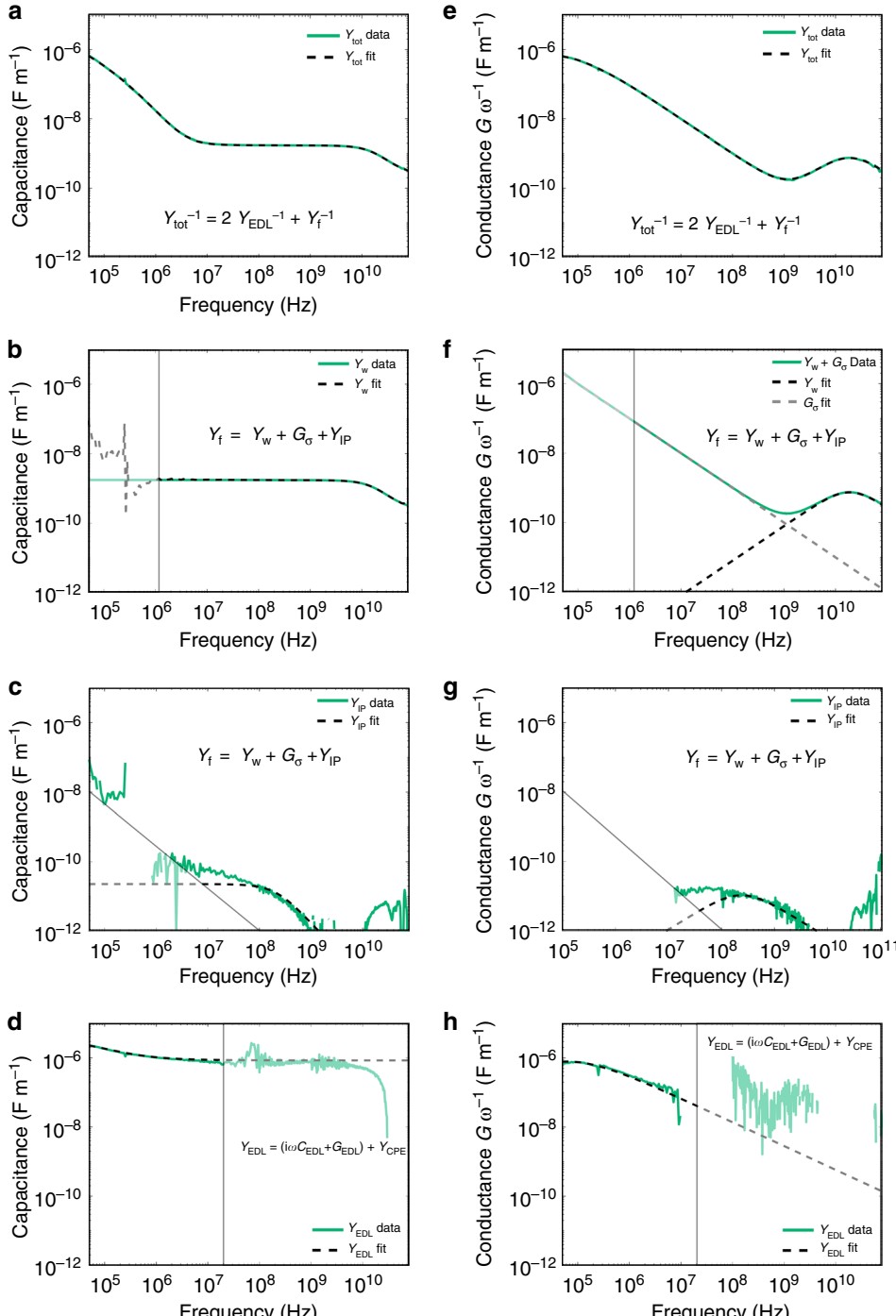

**Fig. 5** Example fitting procedure of $Y_{tot} = G_{tot} + i\omega C_{tot}$ for closed DNA tweezer data. Distributed capacitance (**a**–**d**) and conductance (**e**–**h**) data are green lines in all plots. All dotted lines are fitted equations. Data for the individual circuit components are shown by subtracting all other fitted components from the total distributed capacitance and conductance. **a**, **e** Data and fit of $C_{tot}$ and $G_{tot}$. **b**, **f** Data and fit of $C_w$, $G_w$ (black dotted lines), and $G_\sigma$ (gray dotted line). **c**, **g** Data and fit of $C_{IP}$ and $G_{IP}$. **d**, **h** Data and fit of $C_{EDL}$ and $G_{EDL}$. Lighter shaded regions indicate where the extracted parameters are not well conditioned due to a high signal-to-noise ratio

individual basis. The open tweezer had a lower conductivity than all control measurements, showing the utility of $G_\sigma$ as a parameter for measuring conformational changes and binding events in DNA-based systems.

Other parameters included in the measurement corresponding to the EDL and the ion-pairing relaxation did not yield statistically significant changes for different tweezer conformations. We attributed ion-pairing relaxation observed in the buffer to the solvent-mediated interactions between different buffer

components[32,60]. The addition of closed DNA tweezers did not shift the relaxation magnitude $C_{IP}$ or time constant $\tau_{IP}$ within the error of the measurement (Supplementary Figure 6). The electrical double-layer relaxation fit parameters, $C_{EDL}$ and $\tau_{EDL}$, represent the accumulation of charged species on the surface of the electrode. $C_{EDL}$ measures the capacitive contribution of the EDL, and $\tau_{EDL}$ is the recovery time of the EDL after it is perturbed by the electric field[26,61,62]. These parameters were particularly sensitive to ionic conductivity changes, and adding

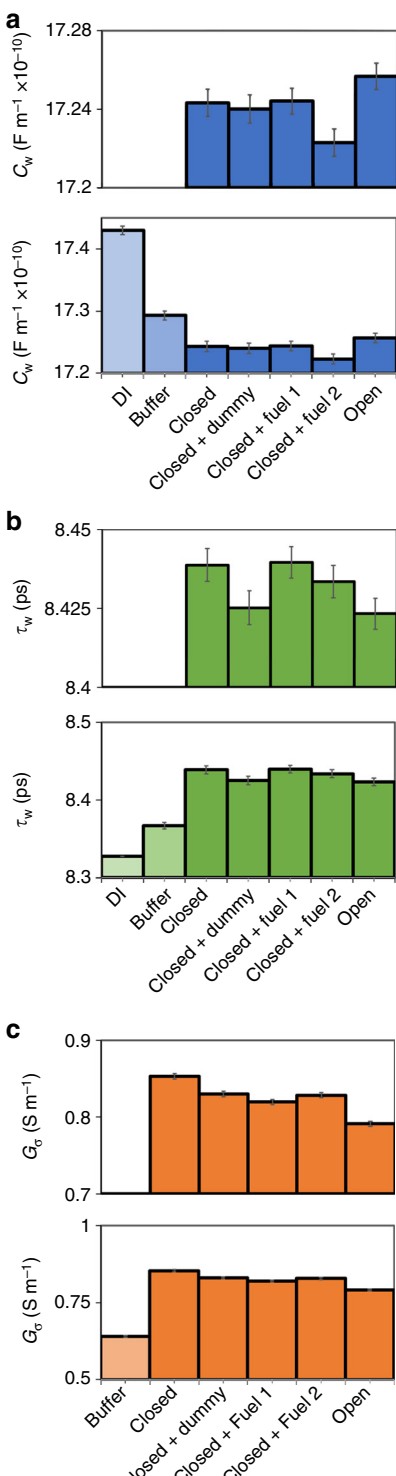

**Fig. 6** Parameters obtained from the fits of broadband capacitance and conductance for tweezer suspensions. **a** Dipolar contribution of water $C_w$. **b** Relaxation time of cooperative water relaxation $\tau_w$. **c** Bulk fluid conductance $G_\sigma$. Error bars represent a 95% confidence interval on the values (top plots provide an enlarged view of the differences between tweezer samples). Closed + loop and Closed + locker correspond to the additions of FUEL 1 and FUEL 2, respectively

DNA increases capacitance $C_{EDL}$ and decreases the relaxation time $\tau_{EDL}$ (Supplementary Figure 5). However, no significant changes were found between any of the tweezer samples. The large time constant $\tau_{EDL}$ of the EDL relaxation in this system

increased the uncertainty in the fit. In future measurements, this uncertainty could be reduced by extending the broadband measurement to lower frequencies. Such an approach could improve the viability of the use of the EDL to probe DNA systems with microwave microfluidics.

## Discussion

While this technique is promising as a method to detect conformational changes in biomolecular systems, several key technical improvements are required to reach its full potential. To realize the promise of this technique and make these measurements accessible to nonspecialist biological laboratories, it is critical to increase the time resolution and develop lower-cost measurements. While lower bandwidth dielectric spectroscopy techniques are commercially available, the broadband nature of these microwave measurements is critical to determining the fitting parameters, including $G_\sigma$ with high accuracy. Fitting the full range of frequencies allows us to capture the effects of ion pairing and the EDL, whose contributions to the electrical signal overlap with the frequency range used to extract $G_\sigma$ (1 MHz–1 GHz). Without accounting for these additional signals, the confidence intervals on the extracted fit parameters would not be small enough to distinguish between open and closed tweezers.

Further studies of biomolecular systems will improve our understanding of the relationship between specific hydration and ion interactions, and the changes in broadband electrical properties that we observe. Our results indicate that more studies are required to elucidate the mechanisms that contribute to the changes in the water relaxation in DNA solutions. For future studies of biomolecules, it is also important to note that the EDL and ion-pairing effects that we detect could both have reasonably been indicators of conformational changes, and may prove to be more sensitive to conformation changes in other systems.

In this report, we demonstrated the first label-free electrical detection of conformational changes in DNA tweezer nanostructures by microwave microfluidics. The extremely wide-frequency range of these measurements allowed us to isolate the effects of the EDL, ionic conductivity, ion-pairing relaxation, and solvent relaxation. We quantified the parameters associated with each of these physical mechanisms and found that the conformational change of the DNA tweezers was most readily detected in the ionic conductivity and the frequency dependence of the water relaxation, while the presence of fuel strands was detected in the water loss relaxation. In particular, the label-free detection of conformational changes on-chip offers opportunities to improve biomolecule characterization by integrating stimuli such as temperature, offering further avenues to measure DNA melt curves and temperature-dependent conformational changes with high sensitivity and high-confidence level (concentration sensitivity of ~20 μg mL$^{-1}$, $p < 10^{-5}$, two-sided Z test with unequal variance, for $G_\sigma$) for nanoliter sample volumes[63,64]. While the measurement techniques developed here are broadly applicable to biological fluids, further microwave measurements and computational studies are necessary to expand the theoretical foundations beyond model systems to interpret the impact of conformational changes on electrical properties. A combination of label-free conformational testing with DNA nanomachines represents a powerful toolbox for understanding the fundamental biological mechanisms, hastening progress in pharmaceuticals, biotechnology, and molecular engineering.

## Methods

**Device fabrication**. The device fabrication for the microwave microfluidics devices is described in detail elsewhere[25,26,65]. Briefly, all devices were co-fabricated on 500-μm-thick fused silica wafers (76.2-mm diameter). We fabricated two separate types of chips: a test chip containing all microfluidic devices (Fig. 2a) and a

reference chip containing bare coplanar waveguide devices for calibration. Metal for coplanar waveguides was deposited by electron-beam evaporation (Ti(5 nm)/Au (500 nm)). All CPW structures were designed and fabricated with 50-μm-wide center conductors, 5-μm-wide gaps, and 200-μm-wide ground planes.

In addition to CPWs of different lengths, we fabricated series resistors, series capacitors, and short-circuit reflects on the reference chip. The series resistor consisted of a 10-μm-wide strip of Ti (1.5 nm)/PdAu (11 nm ± 0.5 nm) resistive material with a measured sheet resistance of ~ 50 Ω. The series capacitor was identical in structure to the series resistor, with the exception that the resistive material was omitted. The short circuit consisted of a region of conductors spanning the ground planes, gaps, and center conductors, connected to a length of transmission lines on either side equal to the length of the thru.

We designed devices with two-layer microfluidic channels consisting of ~50 μm of the SU-8 photoresist, covered with an upper channel layer (~50 μm) of the patterned polydimethylsiloxane (PDMS). The SU-8 microfluidic channels were ~80-μm wide, and were patterned to expose the lengths of CPW directly to the fluid channel (0.50, 0.66, 1.32, 1.98, and 3.13 mm). We chose the CPW gap width and SU-8 channel height so that the electromagnetic fields interact with fluids and SU-8 rather than the PDMS layer. An acrylic press bar screwed into an aluminum chuck was used to clamp the PDMS block to the chip.

**Measurements**. We measured the CPWs with a VNA on a manual microwave probe station (Fig. 2b). We measured the complex scattering parameters (S parameters) as a function of frequency. We acquired 640 frequency points from 100 kHz to 110 GHz on a log-frequency scale, at an AC power of −20 dBm (where 0 dBm corresponds to a power of 1 mW), and with an intermediate frequency bandwidth of 10 Hz. All measurements were performed on a temperature stage controlled to (25 ± 2) °C. After the measurements were performed on the reference and empty test devices, fluid was injected into the channels and held for at least 2 min at zero-flow rate prior to fluid measurements. Each sample measurement lasted ~20–30 min.

We transformed the measured S parameters to distributed circuit parameters for each transmission line segment we measured using the hybrid calibration scheme[26,27,29,66]. Specifically, we performed a two-tier calibration consisting of a reference chip and the fluid-loaded chip. For the first-tier calibration, we measured S parameters for seven different bare CPW lengths (0.420, 1.000, 1.735, 3.135, 4.595, 7.615, and 9.970 mm), a series resistor, a series capacitor, and a short-circuit reflect, all located on the reference chip. The calibration structures used in this work had the same geometry, as described in previous calibrations[67]. We first performed a multiline thru-reflect-line (TRL)[29] calibration to determine the propagation constant of the bare-CPW lines ($\gamma_0$), followed by the series-resistor calibration[66] to compute the capacitance per unit length of the bare CPW section ($C_0$). In the second-tier calibration, we measured four transmission lines, as well as a single short-circuit reflect structure loaded with fluid on the test chip. We then performed multiline TRL calibration and series resistor calibrations with a de-embedding procedure to obtain the propagation constant for the microfluidic channels ($\gamma_{tot}$). The propagation constant for the bare-CPW lines can be written as

$$\gamma_0 = \sqrt{(R_0 + i\omega L_0)(G_0 + i\omega C_0)}, \tag{7}$$

where $\omega$ is the angular frequency and $R_0$, $L_0$, $G_0$, and $C_0$ are the distributed resistance, inductance, conductance, and capacitance per unit length of the bare-CPW lines, respectively, as a function of frequency. We assumed that the conductivity of fused silica was negligible, and the microfluidics fluids over the CPW devices were nonmagnetic. These assumptions allowed us to derive $R_0$ and $L_0$ from the reference chip, and relate them to the propagation constant of a fluid-loaded line:

$$\gamma_{tot} = \sqrt{(R_0 + i\omega L_0)(G_{tot} + i\omega C_{tot})}. \tag{8}$$

The multiline TRL calibration on the microfluidic test chip allowed us to relate the propagation constant of the fluid directly to the capacitance $G_{tot}$ and conductance $G_{tot}$ for frequencies in the range of 1–110 GHz. For frequencies below 1 GHz, where the on-chip CPWs were not long enough to perform multiline TRL, we utilized the series resistor calibration and de-embedded[26–28] our raw measurements to the fluid-loaded portion of the line by accounting for the effect of cables, probes, and the CPW sections leading up to the fluid. For every measurement set, we first measured both air and deionized water in the channels to establish baseline levels for $C_{tot}$ and $G_{tot}$ for known fluid properties.

**Reporting summary**. Further information on experimental design is available in the Nature Research Reporting Summary linked to this article.

## Data availability

The data that support the findings of this study are available from the corresponding author upon reasonable request.

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

## Acknowledgements
The authors would like to acknowledge Ami Thakrar, Aaron Hagerstrom, Derek Houtz, Jasper Drisko, and Nina Popovic for their helpful discussion. The authors would like to thank the National Research Council and the NIST-on-a-Chip Initiative for funding. N.S. acknowledges startup funds from the Arizona State University. This material is based upon work supported by the Air Force Office of Scientific Research under award number FA9550-17-1-0053. N.S. and M.L. would also like to thank Prof. Hao Yan for the use of the AFM. Certain commercial equipment, instruments, or materials are identified in this paper in order to specify the experimental procedure adequately. Such identification is neither intended to imply recommendation or endorsement by the National Institute of Standards and Technology, nor is it intended to imply that the materials or equipment identified are necessarily the best available for the purpose. Official contribution of the U.S. government, is not subject to copyright in the U.S

## Author contributions
A.C.S. made, calibrated, and analyzed microwave microfluidics measurements. M.L. prepared DNA tweezers, performed AFM, and gel characterization. C.A.E.L. fabricated the microwave microfluidic chip. C.J.L., N.D.O., N.S., and J.C.B. contributed to interpretation, analysis, and paper preparation.

## Additional information

**Competing interests:** The authors declare no competing interests.

