## [Peer Review File · Nature Communications]

Reviewers' comments:

Reviewer #1 (Remarks to the Author):

The paper deals with the detection of DNA conformation by an on-chip, label-free and non destructive microwave microfluidics technique.

The proposed approach is original (microfluidics technique for DNA conformation changes detection), the experimental results and consecutive modeling of the paper are relevant, but the discussion needs to be enriched.

It is mandatory to present all the extracted values of the model in a table (in the supplementary data part at least) (values can't be derived from figure 6) with the obtained z-values (no value is given in the paper) and consequently and correctly justify:

- "Strongly indicate" in page 11,
- "Statistically significant" in page 11,
- "High sensitivity" in page 14 (conclusion).

It is also important to comment the fact that only the parameters fitted between 5 and 30GHz (G-mu and Tau-w) correlate with the two solutions of opened and closed DNA and then parts of the spectrum from 50kHz to 5GHz and from 30GHz to 110GHz are useless for the proposed demonstration in this paper.

More importantly: add in the paper the concentrations of :

- DNA tweezers
- Fuel 1,
- Fuel 2,
- Dummy 1,
- Dummy 2.

It will be necessary to also report the concentration's accuracy as well as the Fuels residual concentration after DNA binding.

One hypothesis, which would indeed contradict the last sentence in page 11 and more generally the paper's conclusion, would hypothesize that dielectric changes originated the difference in the single strands' type and concentrations. This point needs to be addressed in the discussion and additional control solutions of buffer + strands (fuels and dummies) at original and residual concentrations need to be measured, added in the figures 3 and 6 and discussed.

Moreover, some points need to be clarified

- End of the first paragraph in page 5 ("This new ability... DNA origami."),
- About "collinearity" in page 10,
- Last sentence of the paragraph "Fitting the fluid ...",
- Y-scale of the figure 1(f) and its significance,
- Fig S3 : the residuals are normalized by which quantity ?

Finally, please:

- Add $\times 10^?$ for the y-scale in fig 6(a),
- Complete some references appropriately.

Reviewer #2 (Remarks to the Author):

Overall comments:

This is an interesting study demonstrating a new approach to probe conformations of DNA nanodevices using microwave microfluidics. The approach has some significant advantages over other methods to probe conformations of DNA nanodevices with the main advantages being the ability to probe without the need for labeling, the relatives of an electrical readout as opposed to an optical readout., and the future possibilities for integration with other lab-on-chip systems. Overall, I think this can be a useful approach for the field. The data are convincing, and I believe the authors did a thorough job in using their model to interpret the data. However, they indicated a key advantage of their approach over other methods such as AFM or TEM is that these other methods are limited to identifying beginning and end states making real-time measurements of intermediate conformations problematic. It is not clear to me that the authors showed this. It seems they are still just probing an end state after actuation rather than that actual actuation process. To clearly demonstrate the advantage of microwave microfluidics over these other methods, it would be important to demonstrate a real-time measurement. I noted a number of other smaller points below.

Specific comments:

- Small point, but I would not say TEM generally requires extensive sample prep. I would agree with this for cryo electron microscopy, but negative stain electron microscopy sample prep is pretty straightforward. I would specify this applies to CryoTEM.
- Researchers have also used high-speed AFM to probe conformational dynamics of origami and other biomolecules. This would be worth mentioning in the literature review. Although I think the time resolution is still limited and experiments are challenging, and it is low throughput (one at a time). But it is worth noting.
- Another drawback that is alluded to but could be made more clearly for some methods like AFM is the potential of surface deposition to perturb solution dynamics.
- I think it is worth pointing out that an electrical readout is more convenient generally than a FRET readouts, which are often done at the single molecule scale to probe dynamics so not the most straight-forward and user friendly readout since it requires a fluorescence microscope (single molecule) and/or fluorometer (bulk).
- In figure 1e, I assume the red and white are taken from unactuated (red) and actuated (white) samples rather than open and closed tweezers that were observed in the same sample? I assume the former given the efficiency seen on the gel, but it would be worth clarifying this point.
- The channel is a little unclear in Figure 2A, can that be labeled or perhaps could a schematic version be included over the overall chip?
- On page 11 at the bottom of the first paragraph, I believe that should reference Fig. S3.
- What is the concentration of the tweezer in solution. Also it would be nice to give the concentrations in molar instead of mol/m³. That is more straightforward to relate to other DNA nanostructure studies.
- It is surprising that open tweezers would increase the amount of free water in solution. I would expect that open tweezers have more surface area accessible for hydration? The authors should comment on possible explanations.
- To verify whether it is the ssDNA in solution that affects the water relaxation, could the authors include a simple control where just ssDNA is added into a channel without tweezers to see if that give similar decrease from buffer? This would identify if those effects are independent. I expect they would be but it would be interesting if they are not. That is also very likely a function of concentration.
- The authors should include AFM images of the tweezers with only the locker and tweezers with only the central loop. Do all tweezers remain closed if only one of those is added? That would be useful to interpret the results in fig. 6.
- Unless I missed this, some details of the measurements are missing, like the concentrations of tweezers and timing, how long were tweezers incubated with actuation strands prior to measurements?
- There is no information about the time resolution of the measurements. How long do the measurements take? What timescale of conformational dynamics or changes could be resolved?

- Also, in terms of making this broadly useful as a platform, it would be nice to have an estimate as to the cost of the device and the measurement system.

Original comment from Reviewer: 1

The proposed approach is original (microfluidics technique for DNA conformation changes detection), the experimental results and consecutive modeling of the paper are relevant, but the discussion needs to be enriched.

We thank the reviewer for noting the originality of our work. We hope that the reviewer agrees that the additional control measurements and added discussion enhance the depth of the manuscript.

Response to Reviewer: 1

1. *The proposed approach is original (microfluidics technique for DNA conformation changes detection), the experimental results and consecutive modeling of the paper are relevant.*

Thank you very much. We worked hard to organize and plan this manuscript and we appreciate such kind feedback.

2. *The discussion needs to be enriched.*

We have incorporated the suggestions from both reviewers to deepen the discussion of the results. We hope that the reviewer agrees that these changes to the manuscript have improved the overall analysis of the results.

3. *It is mandatory to present all the extracted values of the model in a table (in the supplementary data part at least)(values can't be derived from figure 6) with the obtained z-values (no value is given in the paper) and consequently and correctly justify:*
 - *“Strongly indicate” in page 11,*
We have included a p-value ($p < 0.05$) to justify this claim
 - *“Statistically significant” in page 11,*
We have included a p-value ($p < 0.05$) to justify this claim
 - *“High sensitivity” in page 14 (conclusion).*

We have modified the statement:

“In particular, the detection of conformational changes on-chip and label-free offers opportunities to improve biomolecule characterization by integrating stimuli such as temperature, offering new avenues to measure DNA melt curves and temperature-dependent conformational changes with high sensitivity and high confidence level (concentration sensitivity of $\sim 20 \mu\text{g/mL}$, $p < 10^{-5}$ for G_{σ}) for nanoliter sample volumes.^{62,63}”

The reviewer is correct that these values should be explicitly reported. We have included the results in a table in the Supplemental Information. We have additionally included the z-values to qualify each of the phrases specifically suggested by the reviewer.

4. *Its is also important to comment the fact that only the parameters fitted between 5 and 30GHz (G_{μ} and τ_w) correlate with the two solutions of opened and closed DNA and then parts of the spectrum from 50kHz to 5GHz and from 30GHz to 110GHz are useless for the proposed demonstration in this paper.*

While the reviewer is correct that the water relaxation is primarily determined from data between 5 GHz and 30 GHz, to say that the rest of the dielectric spectrum does not contribute to these findings is not quite accurate. First, the strongest indicator of conformational change in the DNA tweezers is the ionic conductivity, which is primarily determined from the 1 MHz to 1 GHz range of the data. Second, fitting the full range of frequencies allows us to capture the effects of the ion pairing and the electrical double layer. Without accounting for these effects, the confidence intervals on our extracted fit parameters would not be small enough to distinguish between open and closed tweezers.

Furthermore, while our findings show that the ionic conductivity is the best indicators of conformational changes in this system, we could not predict these results *a priori*. The frequencies where electrical double layer and ion-pairing effects are present spans the kHz to GHz range and could both have reasonably been indicators of conformational changes.

5. *More importantly: add in the paper the concentrations of: DNA tweezers, Fuel 1, Fuel 2, Dummy 1, and Dummy 2. It will be necessary to also report the concentration's accuracy as well as the Fuels residual concentration after DNA binding. One hypothesis, which would indeed contradict the last sentence in page 11 and more generally the paper's conclusion, would hypothesize that dielectric changes originated the difference in the single strands' type and concentrations. This points needs to be addressed in the discussion and additional control solutions of buffer + strands (fuels and dummies) at original and residual concentrations need to be measured, added in the figures 3 and 6 and discussed.*

This is a great point. We agree with the reviewer that additional controls would serve to identify whether the dielectric changes originated from the difference in the single strands' type and concentrations. To address this point, we have fully repeated all of our experiments and included both the requested controls and additional controls. As we discuss below, we found that the suggested controls did not enable us to identify whether or not dielectric changes originated from the difference in the single strands' type and concentrations, though we believe that our additional controls did enable us to make this determination. Indeed, as the reviewer suggested, we found that the change in capacitance may very well be attributed to the difference in ssDNA. In light of these results, we have revised the manuscript to include our additional set of controls and updated our discussion of the significance of the change in the capacitance associated with the water relaxation.

In more detail, we found that the reviewer's suggested controls (addition of Fuel strands directly into the buffer solution) resulted in much larger changes in the capacitance due to water loss than any of the tweezer samples (where D corresponds to Dummy strands, and F to Fuel strands):

Additionally, we found that despite a decrease in the capacitance due to water with the addition of 9x ssDNA (Buffer to Buffer + 9x D, for example), the capacitance increased with the addition of more ssDNA (from Buffer + 9x D to Buffer + 10x D, for example). Intrigued by this result, we also performed a more extended concentration sweep of fuel strands in buffer:

We found that the ionic conductivity monotonically increased as a function of ssDNA concentration, as we expect for a solution with increasing ionic strength. However, counter to our expectations, the capacitance due to water initially decreased at low ssDNA concentrations, then increased with increasing ssDNA concentration. This repeatable result, though unexpected, may be explained by the concentration dependence of multiple parameters including the charge state of the buffer molecules, the charge density of the DNA strands (see, for example, Zhang et al, *Biophys. Journ.*, 2011), and potentially the addition of other counterions from the solid-phase synthesis of the DNA strands.

Because the impact of additional ssDNA on the water relaxation is not straightforward, we tested additional controls a concentration sweep of fuel added to open tweezers, rather than fuel added to buffer.

We have included this data set as supplemental Fig. S7 to describe the effects of adding additional fuel strands to open tweezers, we and have modified the conclusions of the manuscript accordingly (new manuscript, p. 11 par. 2).

These controls also caused us to modify our conclusions about the frequency of the water relaxation. While our initial measurements showed that the addition of dummy strands reduced τ_w , the concentration sweep of FUEL strands showed that the excess FUEL strands do not affect τ_w beyond the error of our measurements (see Fig. S8). We have modified the manuscript to reflect these findings (new manuscript, p. 12, par. 2). We have also included the uncertainties of sample concentrations in the manuscript (new SI, Table S3).

We note that this does not change the main conclusion of the paper—the change in tweezer conformation is still readily discernable in the ionic conductivity, which cannot be explained by the difference in the ssDNA's type and/or concentration.

6. *Minor points of clarification:*

- *End of the first paragraph in page 5 (“This new ability... DNA origami.”)*

We have corrected this sentence to read:

“This new ability to detect DNA nanostructure changes electrically means that we can use these nanostructures as a model system to study large conformational changes in similar systems. Additionally, we can modify tweezers to detect and amplify small conformation changes in complex biological systems.” (p. 6, par. 1)

- *About “collinearity” in page 10,*

We have corrected this sentence to read:

The inclusion of C_w and $G_w + G_\sigma$ into the fitting model at high frequency was necessary to address the collinearity between G_σ and CPE effects in the model by constraining CPE effects to lower frequencies. The full frequency range of the fit was required to achieve the uncertainties presented here for all extracted fit parameters. (p. 10, par. 2)

- *Last sentence of the paragraph “Fitting the fluid ...”,*

We have corrected this sentence to read:

Residuals of fits for a single Cole-Cole relaxation versus two Debye-type in the GHz frequency range have been reported elsewhere for Mg^{2+} -EDTA buffer and are included for closed tweezers in Fig. S3. [52] (p. 11 par. 1)

- *Y-scale of the figure 1(f) and its significance,*

We have corrected this figure and its caption:

- *Fig S3 : the residuals are normalized by which quantity?*

We have corrected the caption of Fig. S3 to read:

Fig. S3: Residuals normalized to the fit magnitude for C_{tot} and G_{tot} fits (for example, $\frac{C_{tot}-fit(C_{tot})}{fit(C_{tot})}$) including a Debye relaxation for ion-pairing (yellow and black) and not including a Debye relaxation while allowing a Cole-Cole distribution for the water relaxation (orange and gray). (SI page 4)

- Add $*10^?$ for the y-scale in fig 6(a),
We have corrected this scale to read (F / m x 10^{-10}) (p. 33)
- Complete some references appropriately.
We have corrected the references

Original comment from Reviewer: 2

This is an interesting study demonstrating a new approach to probe conformations of DNA nanodevices using microwave microfluidics. The approach has some significant advantages over other methods to probe conformations of DNA nanodevices with the main advantages being the ability to probe without the need for labeling, the relatives of an electrical readout as opposed to an optical readout, and the future possibilities for integration with other lab-on-chip systems. Overall, I think this can be a useful approach for the field. The data are convincing, and I believe the authors did a thorough job in using their model to interpret the data.

We thank the reader for noting the originality and unique advantages of our system. We share the reviewer's interest in seeing the impacts of these types of electrical measurements in lab-on-a-chip systems.

1. *However, they indicated a key advantage of their approach over other methods such as AFM or TEM is that these other methods are limited to identifying beginning and end states making real-time measurements of intermediate conformations problematic. It is not clear to me that the authors showed this. It seems they are still just probing an end state after actuation rather than that actual actuation process. To clearly demonstrate the advantage of microwave microfluidics over these other methods, it would be important to demonstrate a real-time measurement.*

We thank the reviewer for this feedback. We agree with the reviewer that this is an important potential advantage to this technique, and that it has not been demonstrated here. We are actively working on developing higher speed measurements for biofluid systems to demonstrate this important impact. We have moved this potential advantage from the introduction to the discussion as a high-impact avenue for future work.

Because we have not demonstrated real-time measurements in this manuscript, we have removed this claim from the introduction (p. 5, par. 1). We are actively pursuing several of these paths toward faster measurements by incorporating a resonant tracking circuit (see Orloff *et al.* 2015) with an on-chip microfluidic resonator at 10 GHz. These particular experiments, while very interesting, constitute an entirely separate experimental design from the results presented here.

2. *Small point, but I would not say TEM generally requires extensive sample prep. I would agree with this for cryo electron microscopy, but negative stain electron microscopy sample prep is pretty straightforward. I would specify this applies to CryoTEM.*

We thank the reviewer for this feedback, and have edited the manuscript to specify CryoTEM:

Standard characterization methods to detect nanoscale changes in biological systems include cryogenic transmission electron microscopy (cryo-TEM) and atomic force microscopy (AFM).^{1,2} These techniques, however, require extensive and destructive sample preparation (cryo-TEM, for example), or are limited in spatial resolution (AFM, for example). (p. 2 par. 1)

3. *Researchers have also used high-speed AFM to probe conformational dynamics of origami and other biomolecules. This would be worth mentioning in the literature review. Although I think the time resolution is still limited and experiments are challenging, and it is low throughput (one at a time). Another drawback that is alluded to but could be made more clearly for some methods like AFM is the potential of surface deposition to perturb solution dynamics.*

We thank the reviewer for this feedback, and have included references to high-speed AFM measurements and potential drawbacks to AFM measurements in the introduction:

These techniques, however, require extensive and destructive sample preparation (cryo-TEM, for example), are limited in spatial resolution (AFM, for example) and can be perturbative to the state *in situ*. (p. 3, par. 1)

4. *I think it is worth pointing out that an electrical readout is more convenient generally than FRET readouts, which are often done at the single molecule scale to probe dynamics so not the most straight-forward and user-friendly readout since it requires a fluorescence microscope (single molecule) and/or fluorometer (bulk).*

We thank the reviewer for this helpful suggestion, and have included this potential advantage of our measurement in the introduction:

The electrical characterization methods developed here could provide a more user-friendly and high-throughput alternative to optically based measurements such as FRET.⁴⁵ (p. 5, par. 1)

5. *In figure 1e, I assume the red and white are taken from unactuated (red) and actuated (white) samples rather than open and closed tweezers that were observed in the same sample? I assume the former given the efficiency seen on the gel, but it would be worth clarifying this point.*

The reviewer is correct in their interpretation of Fig. 1e, and we have clarified that the two distributions were taken from an actuated and unactuated sample. The new caption for Fig. 1(e) is:

(e) Histograms for the distance between the ends of the tweezer arms based on AFM imaging (red and white histograms are separate samples of closed and open tweezers, respectively). (p. 27)

6. *The channel is a little unclear in Figure 2A, can that be labeled or perhaps could a schematic version be included over the overall chip?*

We thank the reviewer for this suggestion and have included a figure detailing the chip layout (Fig. S6).

7. *On page 11 at the bottom of the first paragraph, I believe that should reference Fig. S3.*

We thank the reviewer for this suggestion, and have adjusted the reference to be Fig. S3

8. *What is the concentration of the tweezer in solution. Also it would be nice to give the concentrations in molar instead of mol/m³. That is more straightforward to relate to other DNA nanostructure studies.*

The reviewer is correct that these values should be specified. We have moved concentrations from the supplemental information into the main manuscript and converted the units to molar. (new manuscript, p. 8)

9. *It is surprising that open tweezers would increase the amount of free water in solution. I would expect that open tweezers have more surface area accessible for hydration? The authors should comment on possible explanations.*

We agree that this result is counterintuitive, and this finding persists in our additional control measurements of open tweezers with varied Fuel strand concentration. We have included a discussion of possible reasons for this effect in our revised manuscript (new manuscript, p. 11-12):

Notably, the magnitude of C_w increased upon the addition of more single-stranded DNA, meaning that there is less bound water in the system overall. This counterintuitive result effect could be due to changes in the buffer, or the concentration-dependent charge density of the DNA itself.^{54,55}

10. *To verify whether it is the ssDNA in solution that affects the water relaxation, could the authors include a simple control where just ssDNA is added into a channel without tweezers to see if that give similar decrease from buffer? This*

would identify if those effects are independent. I expect they would be but it would be interesting if they are not. That is also very likely a function of concentration.

We agree that these experiments would be informative and have performed them. Please see our comment 5 in response to Reviewer 1 for a full description of the additional control measurements.

11. *The authors should include AFM images of the tweezers with only the locker and tweezers with only the central loop. Do all tweezers remain closed if only one of those is added? That would be useful to interpret the results in fig. 6.*

We agree that these AFM measurements would be informative and have included them in the revised manuscript supplementary information (new SI, Fig. S2).

12. *Unless I missed this, some details of the measurements are missing, like the concentrations of tweezers and timing, how long were tweezers incubated with actuation strands prior to measurements?*

The reviewer is correct that this information is missing. We have included these details in the experimental description:

DNA nanotweezers assembly: The DNA strands that constitute each DNA structure were combined in an equimolar ratio in 1×TAE-Mg²⁺ buffer (40 mM Tris, 20 mM acetic acid, 2 mM EDTA and 12.5 mM magnesium acetate, pH 8.0) to reach a final concentration of 0.5 μM per strand. Fuel and dummy strands were added with 10-fold excess and all samples (actuated, unactuated, unactuated with dummy strands) stored 1 to 2 days at 4 °C prior to measurement. (new SI, p. 3)

13. *There is no information about the time resolution of the measurements. How long do the measurements take? What timescale of conformational dynamics or changes could be resolved?*

We agree with the reviewer that the timescale of the measurements is critical to taking advantage of the full possibilities of bulk electrical characterization of biomolecules in solution. Currently, we use a manual probe station to make connections to each device on the chip, and each fluid takes ~20 minutes to measure. We have included the typical measurement time in the Methods section of the revised manuscript (new manuscript, p. 15, par. 1). There are several ways to reduce this overall timescale including developing single device measurements (narrowband or broadband), developing a connectorized package measurement, and using an auto-probing measurement system (see Ma *et al*, IEEE-MTT, 2018).

14. Also, in terms of making this broadly useful as a platform, it would be nice to have an estimate as to the cost of the device and the measurement system.

There are of course two answers to this question: the cost of the system used to develop this technique and the cost of an optimized, deployed system. For the development system used for these experiments, the cost is approximately \$1 million to \$2 million, which includes the probe station, vector network analyzer, chip fabrication and peripherals. For an optimized deployed system, we are working on a comparable platform that costs under \$10 thousand to \$20 thousand. The dramatic decrease in cost leverages connectorized packaged devices which do not require a probe station and using a vector network analyzer with reduced the frequency range. Development could reduce the costs even more as 5G telecommunication technology will drive down the cost of high-frequency test instrumentation.

REVIEWERS' COMMENTS:

Reviewer #2 (Remarks to the Author):

The authors have done a good job addressing my prior comments. In particular, the updated manuscript is significantly clearer to follow the details of the experiments, and the inclusion of new control measurements were critical to improve the results and interpretation of the data. I agree this is an interesting approach that could be quite impactful, but clearly there are challenges remaining that the authors noted in their responses. Namely, the time resolution needs to be sped up and the cost of the system needs to be reduced to make it broadly applicable. I think the authors conclusions are justified, but I would also suggest noting challenges that still need to be addressed to realize the potential. Also, I am not sure if there would be an easy way to integrate this into the conclusions, but I find it quite interesting that there are some counterintuitive results. This could be further explored to perhaps gain new insight into local effects (hydration, ion interactions) that govern DNA nanostructure conformation and dynamics.

REVIEWER REQUESTS:

The authors have done a good job addressing my prior comments. In particular, the updated manuscript is significantly clearer to follow the details of the experiments, and the inclusion of new control measurements were critical to improve the results and interpretation of the data. I agree this is an interesting approach that could be quite impactful, but clearly there are challenges remaining that the authors noted in their responses. Namely, the time resolution needs to be sped up and the cost of the system needs to be reduced to make it broadly applicable. I think the authors conclusions are justified, but I would also suggest noting challenges that still need to be addressed to realize the potential. Also, I am not sure if there would be an easy way to integrate this into the conclusions, but I find it quite interesting that there are some counterintuitive results. This could be further explored to perhaps gain new insight into local effects (hydration, ion interactions) that govern DNA nanostructure conformation and dynamics.

We have included the comments of Reviewer 2 into the new Discussion section in the manuscript:

While this technique is promising as a method to detect conformational changes in biomolecular systems, several key technical improvements are required to reach its full potential. Increasing the time resolution and developing lower cost measurements are both critical to making these measurements accessible to non-specialist biological laboratories. While lower bandwidth dielectric spectroscopy techniques are commercially available, the broadband nature of these microwave measurements is critical to determining the fitting parameters including G_{σ} with high accuracy. Fitting the full range of frequencies allows us to capture the effects of ion pairing and the electrical double layer, whose contributions to the electrical signal overlap the frequency range used to extract G_{σ} (1 MHz – 1 GHz). Without accounting for these additional signals, the confidence intervals on the extracted fit parameters would not be small enough to distinguish between open and closed tweezers.

Further studies of biomolecular systems will improve our understanding of the relationship between specific hydration and ion interactions and the changes in broadband electrical properties that we observe. Our results indicate that more studies are required to elucidate the mechanisms that contribute to changes in the water relaxation in DNA solutions. For future studies of biomolecules, it is also important to note that the electrical double layer and ion-pairing effects we detect could both have reasonably been indicators of conformational changes, and may prove to be more sensitive to conformation changes in other systems.